

# The extended day length promotes earlier flowering of bermudagrass

Mingxia Ji[*], Guangyang Wang[*], Xiaoyan Liu, Xiaoning Li, Ying Xue, Erick Amombo and Jinmin Fu

Coastal Salinity Tolerant Grass Engineering and Technology Research Center, Ludong University, Yantai, Shandong, China
[*] These authors contributed equally to this work.

## ABSTRACT

Day length is a very critical environmental factor affecting plant growth and development. The extension of light application time has been shown to promote flowering in the long-day plant and to shorten breeding time in some crops. However, previous research on the regulation of bermudagrass flowering by light application time is scarce. Therefore, this study investigated the effect of day length on the growth and flowering of bermudagrass by prolonging the light application time in a controlled greenhouse. Three different light application times were set up in the experiment: 22/2 h (22 hours light/2 hours dark), 18/6 h (18 hours light/6 hours dark), 14/10 h (14 hours light/10 hours dark). Results showed that extending the light application time not only promoted the growth of bermudagrass (plant height, fresh weight, dry weight) but also its nutrient uptake (nitrogen (N) and phosphorous (P) content). In addition, daily light integrals were different when flowering under different light application times. Most importantly, under the 22/2 h condition, flowering time was successfully reduced to 44 days for common bermudagrass (*Cynodon dactylon* [L.] pers) genotype A12359 and 36 days for African bermudagrass (*Cynodon transvaalensis* Burtt-Davy) genotype ABD11. This study demonstrated a successful method of bermudagrass flowering earlier than usual time by manipulating light application time which may provide useful insights for bermudagrass breeding.

## INTRODUCTION

Light is not only a source of energy, but it is also one of the most important environmental factors for plant growth and development (*Fukuda et al., 2008*). Light intensity, light quality (spectral qualities of light), and day length are three major factors that have significant influences on plants (*Nadav & Nirit, 2021*). Photosynthesis is directly affected by light intensity, with increased photon flux density at selective wavelengths enabling higher rates of carbon fixation. Light quality influences the synthesis and storage of photosynthetic pigments in leaves, as well as the metabolism of nutrients, and nitrogen (*Shafiq et al., 2021*). It also regulates plant development, photo-morphogenesis, and material metabolism (primary and secondary metabolism) (*Monostori et al., 2018*). Day length is the amount of time that plants are exposed to a predefined light and dark state each day, which

Corresponding author
Jinmin Fu, turfcn@qq.com

influences plant physiological and biochemical reactions to enhance fast growth, and development (*Adams & Langton, 2005*; *Zha & Liu, 2018*). Day length not only plays an important role in inducing external growth of the crop, but also has significant effects on flowering (*Munir et al., 2001*; *Martín et al., 2018*). Plants must shift from vegetative to reproductive development in order to secure their own reproduction in a variety of uncertain surroundings. It depends mainly on the interaction of internal regulation and external factors. The photoperiodic pathway, vernalization pathway, autonomous pathway, and gibberellin pathway are all involved in the regulation of plant blooming.

The purpose of plant breeding is to produce plants with specific characteristics (*Godwin et al., 2019*). Turfgrass breeding is a method to cultivate new varieties and strains of turfgrass with specific traits (such as disease resistance, insect resistance, trampling tolerance, herbicide resistance and stress resistance) and stable inheritance according to the needs of people's production and life (*Casler & Ronny, 1976*). The development of the turfgrass industry is closely related to turfgrass breeding, which is based on traditional breeding methods with the application of newly developed biotechnology. In terms of the remarkable results of breeding, traditional breeding methods usually play a very important role whether in crop breeding, forage breeding or turfgrass breeding, the breeding of breakthrough varieties almost always depends on the repeated hybridization of superior germplasm resources (*Hanna, Raymer & Schwartz, 2013*; *Ahmar et al., 2020*). However, breeding new, advanced varieties for most crops and grasses will take years. The efficacy of agricultural genetic improvement is substantially determined by cycle time, and various ways have been explored to decrease the duration of plant reproductive cycles (*Voss-Fels, Stahl & Hickey, 2019*). Speed breeding, which involves using regulated environmental conditions and extended light application time for early flowering, can shorten the fertility period of turfgrasses and forages and accelerate research through rapid generational progress. *Watson et al. (2018)* reduced the generation time of long-cycle crops such as spring wheat (*Triticum aestivum*) and oilseed rape (*Brassica napus*) to 6 and 4 generations per year, respectively, rather than 2–3 generations under normal glasshouse conditions. *Ghosh et al. (2018)* promoted rapid growth of spring bread wheat (*Triticum aestivum*), durum wheat (*T. durum*), barley (*Hordeum vulgare*), pea (*Pisum sativum*), the model grass, *Brachypodium distachyon* and the model legume, *Medicago truncatula* through speed breeding. This non-transgenic strategy cuts generation time in half, speeds up breeding and other cutting-edge plant breeding procedures, and can meet the genetic gain targets required for our future crops (*Li et al., 2018*; *Wanga et al., 2021*). Turfgrass and forage have a promising future, but there is still a gap in research on rapid breeding of turfgrass and forage.

Bermudagrass has better ornamental color, high density, drought endurance, salt tolerance, wear resistance, and quick reproduction, making it one of the most valued grasses among other warm-season turfgrasses (*Pang et al., 2011*). It's not just one of the world's most popular turfgrasses, but it's also a great plant for soil consolidation and slope protection (*Shi et al., 2014*; *Zheng et al., 2017*). It is also an excellent forage grass due to its rich foliage, soft texture, and good palatability (*Taliaferro, 1995*). Light duration has a great influence on the development of grasses and grains. The growth and development

of tropical forage grasses are influenced by several environmental factors. *Sinclair et al. (2003)* found that extending light application time can promote the growth of 'Tifton 85' bermudagrass. *Esmaili & Salehi (2012)* found that the fresh weight, tiller density, relative water content and reducing sugar content of shoot and root of bermudagrass decreased with the decrease of light duration. In all seed crops, the floral transition is a key developmental switch that determines dry matter production, and flowering time, plant requirements to vernalization are major factors in the climatic adaptation of regions of good germplasm. As turfgrass, it is induced to flower for the purpose of breeding and seed production. Crossbreeding (C. *dactylon* × C. *transvaalensis*) was used to create popular varieties as Tifway, Latitude 36, Northbridge, Tahoma 31, and TifTuf (*Reasor et al., 2016*). Furthermore, because developing genetically stable bermudagrass variants takes a long time, quick breeding is required to enhance the breeding of outstanding germplasm. The objectives of this work were to shorten the flowering time by analyzing the growth and flowering characteristics of bermudagrass under long light application time, in order to accelerate the purification of hybrid progeny and transgenic plants, and to provide a theoretical basis for rapid breeding of bermudagrass.

## MATERIAL AND METHODS

### Plant materials and growth conditions

The plant materials used in this investigation were common bermudagrass (*Cynodon dactylon*) and African bermudagrass (*Cynodon transvaalensis*), designated as A12359 (2n=36) and ABD11(2n=18), respectively (the germplasm resources collected by our research group named A12359 and ABD11). The experimental materials were separated into two identical portions for monitoring growth and blooming. The medium at the experiment was commercial peat soil, and whole bermudagrass stolons were extracted and planted in cylindrical flowerpots (Peilei, Zhenjiang, China). The same number of bermudagrass stolons were planted in each cylindrical flowerpot (six stolons per pot). To remove seeding disparities between individuals, stolons from the mother plant were evenly propagated. Half-strength Hoagland nutrient solution (1/2 HS) was irrigated once a week.

### Treatment

The experiment was carried out in the greenhouse of coastal grass germplasm resources and breeding center of Ludong University on September 2020. The location is 127 m above sea level, with geographical coordinates of N37.53° (latitude) and E121.36° (longitude). The entire experiment lasted three months, from September to December 2020. All experimental materials were grown on plant shelves (the wire beneath the shelves was breathable and temperature-permeable) in the controlled greenhouse and were supplemented by sodium metal halide lights (RVP350; PHILIP Shanghai China) positioned at a height of 1.8 m from the shelves. Three light application times, namely 22/2 (22 h light/2 h dark), 18/6 (18 h light/6 h dark), and 14/10 (14 h light/10 h dark) were used. Each light application condition had the same bermudagrass (three replicates per material, one replicate per pot). The supplementary light was set to automatically turn off for two hours from 9 pm to 11 pm to achieve a 22/2 light application time. Under the same conditions, 18/6 and

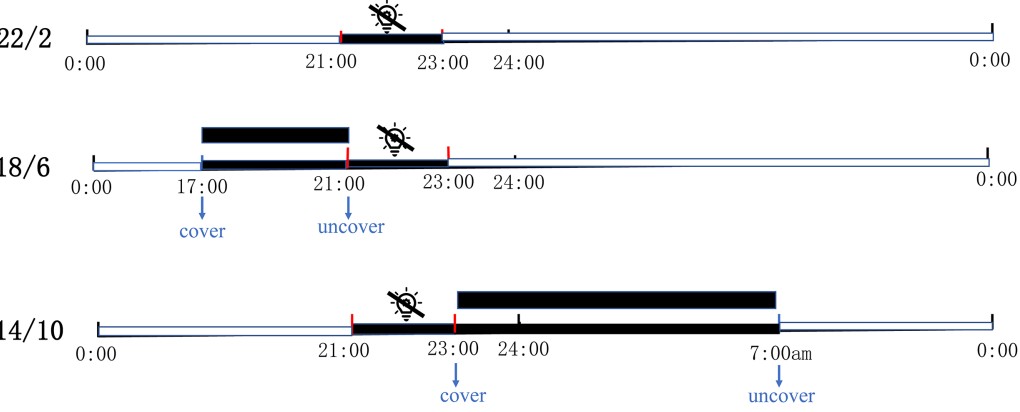

**Figure 1** **Schematic diagram of three artificially designed light application times.** The control of light application time was achieved in controlled greenhouses by supplementary lights. The supplementary light was automatically turned off for two hours from 9 PM to 11PM every day to achieve a 22/2 Light application time; 18/6 Light application time was covered by paper boxes at 5 pm and uncovered at 9 pm; 14/10 Light application time was covered at 9 pm and uncovered at 7 am the next day. Black square shows cover with box (complete darkness).

14/10 were covered by paper boxes at regular intervals and guaranteed complete darkness. The 18/6 light application time was covered at 5 pm and uncovered at 9 pm; 14/10 light application time was covered at 9 pm and uncovered at 7 am the next day (Fig. 1). The greenhouse had a fixed relative humidity and atmospheric pressure of 60% and 101 kpa respectively. In the experiment, three light application times were treated under the same temperature and light intensity. A temperature and light recorder (WS1PROG; ubibot, Dalian, China) was used to record the temperature and light intensity in greenhouse (Fig. S1).

### Recording inflorescence number and growth indicator

Every two weeks, growth indices such as plant height, branch number, and biomass (fresh weight, dry weight) were assessed under various light application times. The growing material was clipped every two weeks. At the specified times, each pot was harvested manually using a hand shear at a consistent height of 10 cm above the soil surface. Every two weeks, the plant height in each pot was measured from the soil surface to the plant's topmost tip (using the standard scale), and the branches were counted. After mowing, the fresh weight was weighed using an electronic balance. The samples were then dried for 30 min at 105 °C in a forced-draft oven (DHG-9140A; Shanghai, China) before being oven-dried for 72 h at 75 °C until they reached a consistent weight. After drying the samples, the dry weight (DW) was weighted. The above-ground (shoots) and subsurface sections (roots) of bermudagrass were separated at the end of the experimental treatment. After carefully washing the tissues with deionized water, the fresh weight (FW) of the shoots and roots was weighted, and the dry weight (DW) was weighted after drying the samples. We counted the number of headings and flowering every two days after the first inflorescence developed until the number of headings and flowering were steady.

### Determination of nitrogen and phosphorus content

To assess the nutritional value, shoots were pulverized and weighed around to 0.1 g, placed in a desiccating tube, mixed with 5 ml of 95% concentrated sulfuric acid ($H_2SO_4$) and digested in a graphite digestion apparatus (SH220N; Jinan Hanon, Shandong, China). Then, using a chemical automated analyzer, the nitrogen (N) and phosphorus (P) content was determined (SmartChem 200; AMS Alliance, Guidonia, Rome, Italy).

### The average photosynthetic photon flux density (PPFD) and daily light integral (DLI)

Based on the temperature and light intensity recorded in Fig. S1, we calculated the average photosynthetic photon flux density (PPFD) for each treatment and temperature, and the average daily PPFD for 22/2, 18/6, and 14/10 h application times were: 98.93 $\mu$mol m$^{-2}$s$^{-1}$, 83.54 $\mu$mol m$^{-2}$s$^{-1}$, 66.29 $\mu$mol m$^{-2}$s$^{-1}$, respectively. The average temperature was 20 °C. Furthermore, we integrated the total light energy received by the material under three light application times. The interval of DLI for different light application times was one week.

### Data analysis

For data processing, Microsoft Excel was utilized. For date visualization, Sigmaplot12.3 was used. Originlab was used for greenhouse temperature and light mapping. SPSS ver. 22.0 software was used to conduct statistical analysis (Microsoft Corp). The noteworthy differences between various light application times were analyzed using One-way analysis of variance (ANOVA) ($P < 0.05$). In this study, two ploidy bermudagrass were chosen as the representative to reflect their response characteristics to photoperiod, so only the effect of photoperiod on them was emphasized and statistically analyzed, and compared under the same conditions of the same species in the same week. The differences caused by light application time were marked by letters (a, b, c). Otherwise, Two-way analysis of variance (ANOVA) was employed to examine the effects of 'Light application time', 'Week' and 'Light application time * Week' ($p < 0.05$) (Table S1).

## RESULTS

### Effect of different light application times on the growth of bermudagrass

The result was presented in Fig. 2, where the charts depicted the change in plant height during grass mowing. For common bermudagrass (A12359), the interaction between Light application time and week was significant (Table S1), and different light application times had a substantial effect on plant height, under longer light application times 22/2 and 18/6 h being higher than that under shorter light application time 14/10 h (Figs. 2A and 2B). In addition, the plant height of common bermudagrass, A12359 tended to rise in early stages (2–4 weeks) before it stabilized in the later stages (Fig. 2A). For African bermudagrass (ABD11), there was significant interaction between light application time and week (Table S1). Moreover, plants with longer light application times (22/2 and 18/6 h) were taller than those with shorter (14/10 h), and the difference was significant. It was worth noticing that the plant height of African bermudagrass ABD11 did, however, show a consistent upward

tendency (Fig. 2B). The pattern of plant height variation suggested that frequent mowing contributes to the growth of African bermudagrass.

The biomass (fresh weight) was measured every two weeks, as illustrated in Figs. 2C and 2D. There were considerable changes in fresh weight while grown under different light application times. And for plants grown under longer light application times *viz* 22/2 and 18/6 h were found with higher fresh weight compared to plants under shorter light application time 14/10 h (Figs. 2C and 2D). For common bermudagrass (A12359), the interaction between light application time and week was significant (Table S1), and A12359's fresh weight in the 22/2 and 18/6 h conditions differed significantly from that in the 14/10 h condition (Fig. 2C). Further, the maximal biomass for A12359 was attained in week twelve (Fig. 2D). For African bermudagrass (ABD11), the interaction between light application time and week was significant (Table S1), however, there was no significant difference in fresh weight between 18/6 and 14/10 h treatments, while 22/2 h was considerably different from the other two groups (Fig. 2D). The maximal biomass for ABD11 was attained in week eight (Fig. 2D).

The study discovered that variations in dry and fresh weights of A12359 and ABD11 essentially followed the same pattern. The dry weight exhibited considerable changes under different light application times. Dry weight was greater when plants were grown at a longer light application times (22/2 and 18/6 h) compared to shorter light application time (14/10 h) (Figs. 2E and 2F).

The fresh weight and dry weight of the bermudagrass shoots and roots were evaluated in order to determine the influence of light application time on their biomass. As predicted, longer light application times resulted in higher biomass of both shoots and roots than shorter light application times (Figs. 2G and 2J). The fresh weight of common bermudagrass (A12359) shoots and roots exhibited significant variations between the three light application times (Fig. 2G), while the dry weight at longer light application time 22/2 h showed significant differences from the other two groups (Fig. 2H). Furthermore, the fresh weight of shoots of African bermudagrass (ABD11) at 22/2 h was significantly different from the other two groups, but the fresh weight of roots was not significantly different among the three light application times (Fig. 2I). Nevertheless, dry weight of its roots, at 22/2 h was significantly different from the other two light application times (Fig. 2J).

## Effect of different light application times on nutrient absorption of bermudagrass

For A12359, the effect of different light application times on N and P content was significant, and the interaction between light application time and week was significant (Table S1). And plants at longer light application time, N and P content was higher than the shorter (Figs. 3A and 3C). Under the light application time 22/2 h, A12359 achieved its maximum N and P at week 12. For ABD11, the effect of different light application times on N and P content was significant, and the interaction between light application time and week was significant (Table S1). Moreover, the N and P content at the longer light application time

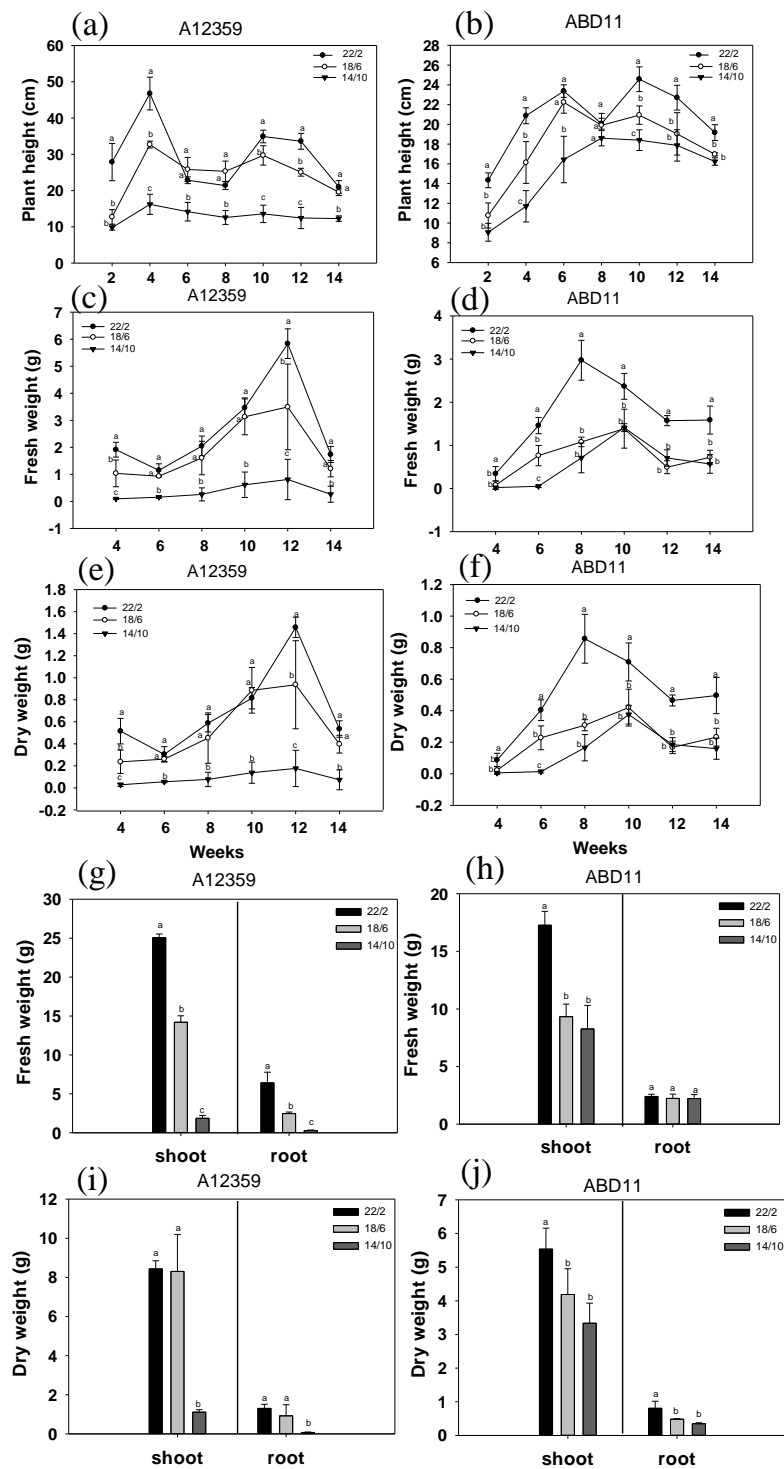

**Figure 2** **Plant height (cm), fresh weight (g) and dry weight (g) of common bermudagrass (A12359) and African Bermudagrass (ABD11) grown in three light application times (22/2, 18/6 and 14/10 hrs light/dark).** The graph depicts the plant height, fresh weight and dry weight, of A12359 and ABD11. (A) A12359 and (B) ABD11 Changes in plant height every two weeks under three light application times.

**Figure 2 (…continued)**
(C, D) Biomass (fresh weight) of A12359 and ABD11 measured every two weeks at three light application times. (E, F) Dry weight of A12359 and ABD11 measured every two weeks at three light application times. (G, H) At the end of the experiment, the fresh weights of shoot and root of A12359 and ABD11 were measured under three light application times. (I, J) At the end of the experiment, the dry weights of shoot and root of A12359 and ABD11 were measured under three light application times. Fresh and dry weights are in grams per pot. Each material is designed to be replicated three times. According to the SNK test at $P < 0.05$, different lowercase letters indicate significant differences between the three light application times at the same time.

22/2 h showed significant differences from the other two groups (Figs. 3B and 3D). Under the 22/2 h, ABD11 reached its maximum N and P at week 8 (Figs. 3B and 3D).

**Effect of different light application times on flowering of bermudagrass**
Branch numbers were greater when plants were grown at longer light application times (22/2 and 18/6 h) compared to the shorter time (14/10 h) (Figs. 4A and 4B). For common bermudagrass (A12359), the effect of different light application times on branch number was significant, and the interaction between light application time and week was significant (Table S1). And branch numbers were greater when plants were grown at longer light application times (22/2 and 18/6 h) compared to shorter (14/10 h) (Fig. 4A). The common bermudagrass (A12359) did not have a propensity to increase its branching number at 14/10 h (Fig. 4A), indicating that A12359 could only branch when it reached a particular daylight level, and the daylight length 14/10 h was insufficient for its branching (Fig. 4A). However, the similar pattern of branching number expansion was seen in both 18/6 and 22/2 h light application times (Fig. 4A), suggesting that 18/6 could be adequate for branching for A12359. Furthermore, by week 12, the common bermudagrass (A12359) reached its maximum number of branches (Fig. 4A). For African bermudagrass (ABD11), the effect of different light application times on branch number was significant, and the interaction between light application time and week was significant (Table S1). The number of branches with a light application time of 22/2 h was greater than that of 18/6 and 14/10 h, and the difference was significant. In African bermudagrass (ABD11) branching number increased as daylight duration rose, and the link between daylight length and branching number showed a linear change (Fig. 4B).

The blooming number of bermudagrass displayed an inverse curve trend in response to different light application times (Figs. 4C and 4D). For common bermudagrass (A12359), the first to flower at the lengthy light application time, 22/2 h. It took 44 days under longer light application time 22/2 h, 63 days under 18/6 h, and 85 days under short light application time 14/10 h from planting to flowering, *i.e.*, the longer light application time shortened common bermudagrass flowering time by 41 days. For African bermudagrass (ABD11), the first to flower was produced at longer light application times, 22/2 h and 18/16 h. ABD11, on the other hand, took 36 days from planting to blooming under light application time 22/2 h, 36 days under 18/6 h, and 95 days under short light application time 14/10 h. These results suggested that extending light application time might promote flowering in common bermudagrass and African bermudagrass.

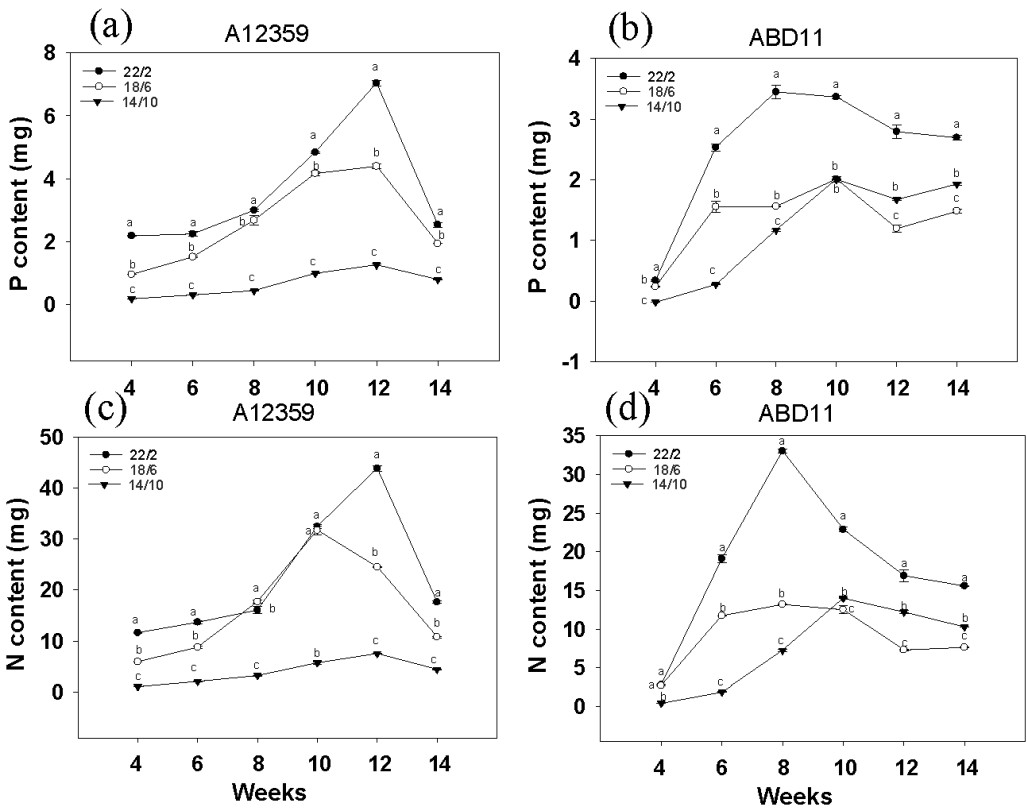

**Figure 3   Nitrogen (N) and phosphorus (P) concent of common Bermudagrass (A12359) and African Bermudagrass (ABD11) grown in three light application times (22/2, 18/6 and 14/10 hrs light/dark).** The graph depicts the N and P content of A12359 and ABD11. Fresh samples from each mowing were dried and utilized to determine the nitrogen (N) and phosphorus (P) levels. The N and P content measured was accumulated over two weeks. (A, B) Changes of P content in A12359 and ABD11 under different light application times. (C, D) Changes of N content in A12359 and ABD11 under different light application times. Each material is programmed to have three duplicates. According to the SNK test at $P < 0.05$, different lowercase letters indicate significant differences between the three light application times at the same time.

## Daily light integral for different light application times

The daily light integral curves for the three light application times almost linearly increased with the extension of the treatment time, as shown in Fig. 5. Moreover, the daily light integral of 22/2 h was the most in the whole experiment. For material A12359, the long light application time 22/2 h flowered at week 6.43, the accumulated daily light integral was approximately 382 mol m$^{-2}$ d$^{-1}$, the 18/6 h treatment flowered at week 9.14, the accumulated daily light integral was approximately 482 mol m$^{-2}$ d$^{-1}$, and the 14/10 h treatment flowered at week 12.43, the accumulated daily light integral was approximately 582 mol m$^{-2}$ d$^{-1}$. The longer the light application time, the earlier the flowering, however, the longer light application time accumulates the least amount of daily light integral at the first flowering. A12359 showed different daylight integrals when flowering under different light application times, while ABD11 showed roughly similar daylight integrals when flowering under 22/2 and 18/6 h.

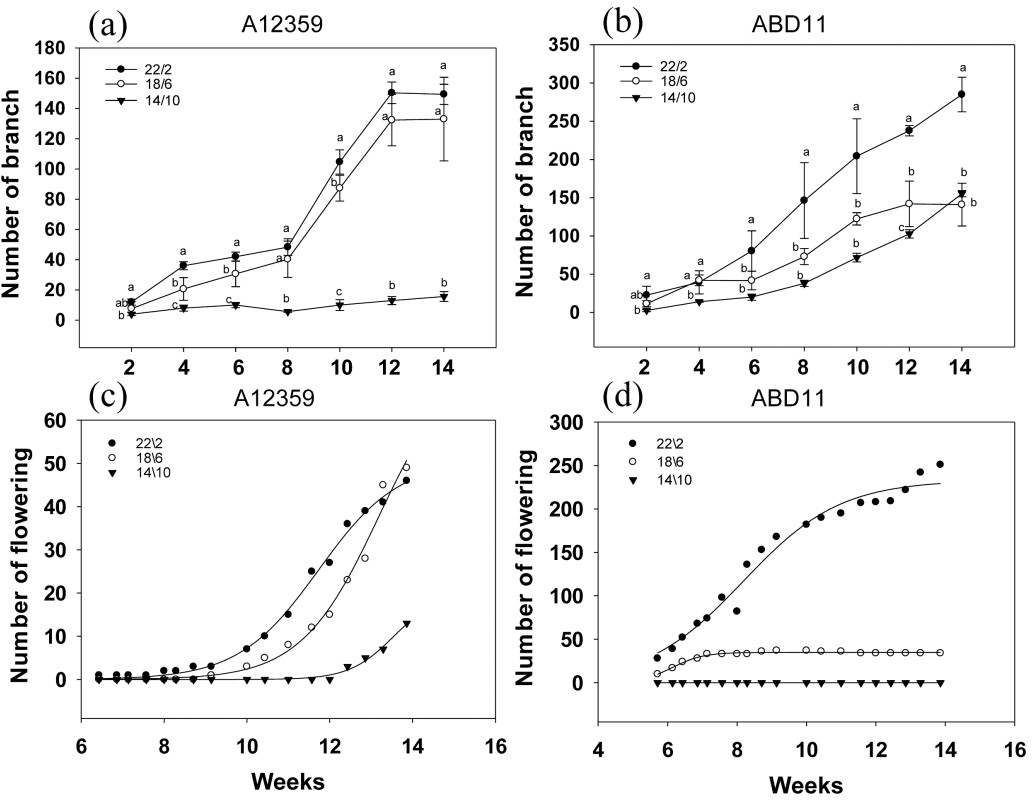

**Figure 4** **Branching and flowering of common bermudagrass (A12359) and African bermudagrass (ABD11) grown in three light application times (22/2, 18/6 and 14/10 hrs light/dark).** The graph depicts the blooming and branching of A12359 and ABD11. Every two weeks, the number of branches was tallied, and every two days, the number of blossoms was counted. (A) Branch number variation of A12359 under different light application times. (B) Branch number variation of ABD11 under different light application times. (C) Changes of flowering number of A12359 under different light application times. (D) Changes of flowering number of ABD11 under different light application times. Each material is designed to be replicated three times. According to the SNK test at $P < 0.05$, different lowercase letters indicate significant differences between the three light application times at the same time.

# DISCUSSION

Light duration has a significant effect on the developmental pattern of grasses and cereals. It has been discovered that increasing light application time might encourage the growth of bermudagrass biomass. The fresh and dry weights of bermudagrass, for example, rose considerably in light application times 22/2 and 18/6 h as compared to the shorter light application time 14/10 h (Figs. 2C–2F). This observation was consistent with previous studies of *Hay (1990)* in case of dry-matter production of grass species like high-latitude types of *Phleum pratense* (both vegetative and reproductive), *Poa pratensis* (vegetative), and Bromus inermis (vegetative). *Thomas, Paul & Jeffery (2001)* found that the yields of the three species (*Paspalum notatum* Flugge; *Cynodon dactylon* L.; *Cynodon nlemfuensis* Vanderyst) under long light application time were higher than those under natural daylight. But the growth of Florakirk bermudagrass and Florona stargrass was not as pronounced as

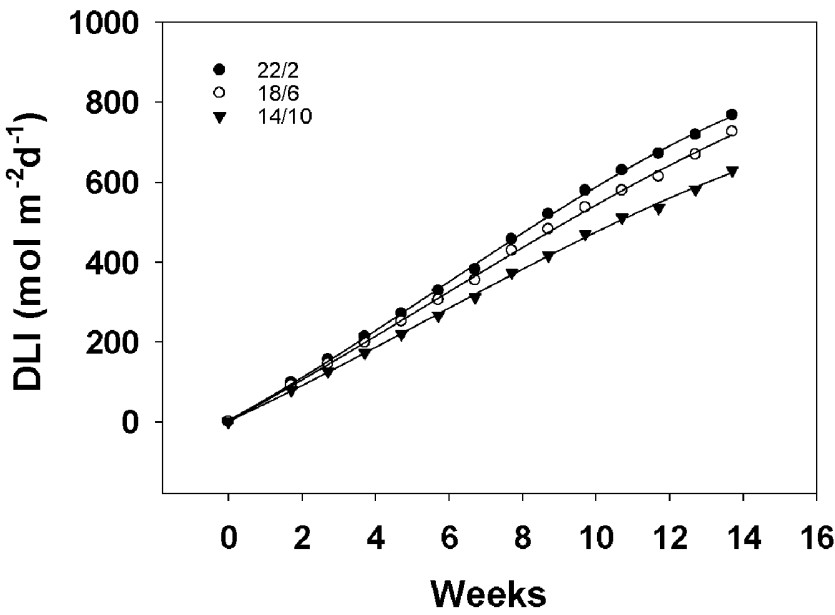

**Figure 5** **Daily light integral (DLI) for different Light application times.** This graph depicts the daily light integral for three different light application times. We integrated the total light energy received by the material under three light application times. The interval of DLI for different light application times was one week.

Tifton 85 bermudagrass or Pensacola bahiagrass. Grasses that were not sensitive to daylight also showed decreased yields under short light application time. *Trenholm et al. (1998)* found that the shoot growth of the 'Flora Dwarf' bermudagrass had a 41% decrease in short light application time compared to long light application time. However, *Marousky et al. (1992)* reported that the shoot growth of 'Tif Dwarf' bermudagrass did not increase significantly under long light application time. It is common that long light application times can promote plant growth, but the effects vary among species.

Flowering transition is a critical feature in plant growth that signifies the end of the vegetative phase and the start of the reproductive state (*Quiroz et al., 2021*). Prior to the flowering transition, plants undergo a transition from juvenile to adult as they reach their flowering capacity under adaptive induced environmental conditions. And light application time is one of the main environmental stimuli. Most plants have their own biorhythm clock that can sense changes in the external light environment, so the duration of light can significantly affect the flowering of plants (*Kobayashi & Weigel, 2007*). Flowering of plants can be advanced or delayed by artificially changing the length of the light application time. In our investigation, it was found that prolonged light application time in the controlled greenhouse might encourage the bermudagrass to bloom earlier, with the bermudagrass blooming first under the circumstance of a longer light application time 22/2 h (Figs. 4C and 4D). This finding was congruent with that of *Arabidopsis thaliana*, a characteristic long-day plant that blooms sooner on long days and slows blooming on short days (*Samach, 2000*; *Suárez-López et al., 2001*; *Fornara, Montaigu & Coupland, 2010*; *Ye et al., 2021*). The

correlation between the response to light application time and climatic conditions of forage origin appears to be complex. *Heide (1982)* found that the requirements of light application time of "Timothy" varied according to the geographic origin of the cultivar. Therefore, it needs to be further verified whether other germplasm of bermudagrass can promote flowering under long light application time. However, light intensity seems to be particularly important during the transition from vegetative to reproductive stages. In general, an increase in the light intensity or daily light integral increases biomass and flower developmental rate. The extension of light application time also increased the total light energy irradiated on the plant, and the daily light integrals were different when flowering under different light application times (Fig. 5). Exposure to high levels of light intensity can hasten time to flowering (*Munir et al., 2004*). For example, *Achillea millefolium* flowered after 57, 45, and 37 days when grown under 100, 200, or 300 $\mu$mol m$^{-2}$s$^{-1}$ respectively (*Zhang et al., 1996*). *Arabidopsis* plants also flower earlier under high light intensity. The use of longer light application time to boost plant development has long been investigated; *Velez-Ramirez et al. (2010)* undertook a thorough survey of the literature on this issue published in the previous 90 years (*Velez-Ramirez et al., 2010*), including spring wheat (*Triticum aestivum*), barley (*Hordeum vulgare* L.), peas (*Pisum sativum* L.), radish (*Raphanus sativus*), alfalfa (*Medicago sativa*), flax (*Linum usitatissimum*), and arabidopsis (*Arabidopsis thaliana*). The current attention to the molecular mechanism of flowering regulation is mainly focused on model plants such as Arabidopsis, rice and wheat, and there are few studies on flowering in forage grasses and turfgrasses, so the applicability of the current flowering mechanism in the plant world needs to be further explored.

In addition, for plant introduction and breeding, extending the light in the controlled greenhouse and artificially changing the light application time are crucial. The technology of speed breeding, which allows for rapid generational growth, has been tweaked to generate up to six generations of wheat each year. As a result, it's a useful tool for cutting the breeding cycle (*Alahmad et al., 2018*; *Zheng et al., 2013*). It takes years of thorough screening and testing of warm-season turfgrass varieties during seed propagation to uncover superior genotypes that persist and are considerably superior to the present varieties (*Hanna, Raymer & Schwartz, 2013*). Several molecular techniques have been developed to expose the diversity of crop natural germplasm resources (*Farsani et al., 2012*). Rapid generation evolution to purity following crossing in a breeding setting will boost genetic gain of essential features and enable for more rapid development of better varieties through the breeding program (*Bonos, Clarke & Meyer, 2006*). As a result, we proposed a rapid generation system for bermudagrass in the greenhouse with additional supplemental light, extending the daylight length to 22/2 h, to accelerate plant development and thus shorten the breeding time, and it was successful in reducing the flowering time of common bermudagrass to 44 days and that of African bermudagrass to 36 days. It has been demonstrated that African bermudagrass has a high genetic diversity and may be utilized to improve intraspecific and interspecific breeding (*Kenworthy et al., 2006*). To speed progress, it is critical to identify the flowering thresholds of African bermudagrass and common bermudagrass.

This study shortened the flowering time by increasing the light application time and provided a theoretical basis for speed breeding of bermudagrass. Furthermore, fast breeding

may be integrated with molecular biology, such as genomics, high-throughput sequencing, and genome editing, to increase the speed and precision of developing excellent varieties.

## CONCLUSION

This study was carried out in the controlled greenhouse in order to explore the influence of enhanced daylight length on the development and flowering of bermudagrass. The changes of plant height, fresh weight, dry weight and nitrogen and phosphorus content were analyzed and concluded that long light application time promoted the growth and nutrient uptake (N and P content) of bermudagrass. In addition, daily light integrals were different when flowering under different light application times. Longer daylight length 22/2 h (22 h light/2 h dark) clearly enhanced the development and flowering and eventually reduced the flowering period to 44 days for common bermudagrass and 36 days for African bermudagrass. Therefore, the breeding time can be shortened by increasing the light application time. This study provides a theoretical basis for speed breeding of bermudagrass.

## ACKNOWLEDGEMENTS

We thank Prof. Jinmin Fu for the valuable advice on the design of the experiments.

### Funding

This work was supported by the National Key Research and Development Program of China (2019YFD0900702) and Agricultural Variety Improvement Project 366 of Shandong Province (2019LZGC010). The funders had no role in study design, data collection and analysis, decision to publish, or preparation of the manuscript.

### Grant Disclosures

The following grant information was disclosed by the authors:
National Key Research and Development Program of China: 2019YFD0900702.
Agricultural Variety Improvement Project 366 of Shandong Province: 2019LZGC010.

### Competing Interests

The authors declare there are no competing interests.

### Author Contributions

- Mingxia Ji performed the experiments, analyzed the data, prepared figures and/or tables, authored or reviewed drafts of the article, and approved the final draft.
- Guangyang Wang conceived and designed the experiments, authored or reviewed drafts of the article, and approved the final draft.
- Xiaoyan Liu performed the experiments, analyzed the data, prepared figures and/or tables, and approved the final draft.

- Xiaoning Li performed the experiments, prepared figures and/or tables, and approved the final draft.
- Ying Xue analyzed the data, prepared figures and/or tables, and approved the final draft.
- Erick Amombo analyzed the data, prepared figures and/or tables, and approved the final draft.
- Jinmin Fu conceived and designed the experiments, authored or reviewed drafts of the article, and approved the final draft.

## Data Availability

The raw data is available in the Supplemental File.

## Supplemental Information

Supplemental information for this article can be found online at http://dx.doi.org/10.7717/peerj.14326#supplemental-information.

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

## FURTHER READING

**Choudhury S, Panda P, Sahoo L, Panda SK. 2013.** Reactive oxygen species signaling in plants under abiotic stress. *Plant Signaling & Behavior* **8(4)**:e23681 DOI 10.4161/psb.23681.

**Szymańska R, Ślesak I, Orzechowska A, Kruk J. 2017.** Physiological and biochemical responses to high light and temperature stress in plants. *Environmental and Experimental Botany* **139**:165–177 DOI 10.1016/j.envexpbot.2017.05.002.