# Peer review of "The extended day length promotes earlier flowering of bermudagrass"

_PeerJ, doi:10.7717/peerj.14326_

## Round 0.1 · original submission · Major Revisions

Dear Authors, You need major improvement in the manuscript. Please revise the manuscript according to the reviewers' comments.

·

Basic reporting

A) English is mostly understandable, however, need to clear several sentences as mentioned in the body of manuscript.
B) Literature references and background information are adequately sufficient.
C) Article structure, figures, tables are appropriate.
D) Results are described as per objectives.

Experimental design

A) Not clear which design of experiment was followed.
B) Rests are adequate.

Validity of the findings

A) Graphs/charts are sufficient to represent the results of the experiment.
B) Outcome of the experiment has satisfied the objectives.
C) Discussion need to be more concise, relevant and specific. Need to discuss according to experimental objectives, not beyond that.

Additional comments

The manuscript has been prepared based on the objectives set. However, it needs a careful consideration to check the structure and sentences wherever necessary.

Need to follow the journal's prescribed format to prepare the manuscript, specially for Reference section. This section needs lots of correction and careful observations.

Follow the corrections, suggestions and queries provided in the body of the manuscript by review pan.

Reviewer 2 ·

Basic reporting

I have read the paper entitled ″Long daylength promotes the flowering transformation of bermudagrass″ written by Ji et al. I appreciate the authors for their efforts working with an interesting theme. However, the quality of the paper beyond the standard for publication. There are some major concerns (listed below) and therefore, need substantial revision. Plenty of typing errors existed and English language must be greatly improved to clearer the sentences at several places in the manuscript (highlighted with yellowish marks).
In concluding remarks, the research finding lacks novelty, remain errors in experimental design and data analysis, inadequate data visualization and requires proper background info and discussion with relevant literatures. The language and structure of the manuscript need considerable improvement.

Experimental design

The research is original and within the aims and scope of the journal.
Title is not precise and informative according to the findings. ″Flower transformation″ words/terms need to be rephrased.
Introduction lacks relevant literatures especially the information regarding bermudagrass is scanty. Introduction focused more on plant breeding which is the prospects of the study but the background info in relation to morphological and physiological responses of bermudagrass to daylength or light intensity is ignored. Surprisingly, study objectives were not clearly mentioned at the end of the introduction part, rather a mixture of methods and conclusion was summarized.
The methodology of the study is not appropriate and clear. The authors maintained three light environments with the same light intensity. It is hard to understand the amount of light received by each photoperiodic condition in Figure 2b (very unclear figure). If used same light intensity in each treatment, then the daily light integral (DLI; light intensity*daylength) was different for each photoperiodic condition that would mainly influence the growth and flowering. However, photosynthetic photon flux density (PPFD, µmol m-2s-1) instead of Lux should be used incase of plants growth and development studies with light. Please clarify the meaning of three biological replicates (L108). A single pot considered as a single replicate?? How many stolons per pot? How many pots per treatment? What is the reason for the designation of common bermudagrass and African bermudagrass as A12359 and ABD11?
The plants were grown under three photoperiodic conditions maintaining individual metal halide lamps (L 124). Then why did the lights turn off at the same time? Why not according to the time schedule? Keeping the plants completely covered with boxes (L126-129) would alter the microclimate like temperature and relative humidity. Therefore, the temperature and relative humidity were not identical among the treatments and mixed up certainly. Please clearly mention the set temperature, relative humidity and atmospheric pressure inside the greenhouse (and possibly in each treatment). I assumed the atmospheric pressure (calculated with relative humidity and temperature) greatly differed among the treatments and affect the plants growth accordingly. A photograph of experimental (treatments) set up is obligatory. Growth analysis parameters like crop growth rate (CGR), relative growth rate (RGR) etc. data should be calculated and included.
Data were analyzed using one-way ANOVA (L161) although the data were collected at different time intervals and in two grass species. Figure 3 showed the differences between the species without considering statistical analysis (two factorial). Interaction effects of photoperiods and weeks are missing due to one factorial data analysis.

Validity of the findings

Results showed redundancy. Figures need extensive improvement with appropriate statistical information. Figure titles are not self-explanatory. Difficult to understand whether the lowercase lettering (significance) within the treatments or among the time intervals. The comparison between two species were highlighted without considering statistical significance between them (commented above). Figure 2g-j must be shown with bar graph comparing two species in the same graph. Figure 2c and 2g are confusing and unit must be clarified (g per plant or pot??).
Discussion lacks relevant literatures and consists with other crops escaping the target crop. The influence of daylength or daily light integral (DLI) on biomass production and flowering in bermudagrass is completely ignored which is the key focus of the study. The physiological mechanisms behind the higher accumulation of biomass and early flowering in bermudagrass under longer photoperiod need to be properly discussed. Rather, a large part of discussion consisted of plant breeding methods and technologies which have not been worked or considered in this study. However, the longer photoperiod maintained by artificial lighting is expensive and energy cost will be a concern. Prolonged photoperiod like 22 h might also cause leaf chlorosis and other physiological disorders.
Conclusion reflects objective and key finding but missing perspectives.
Reference format is not unique. Many typing errors, misspelling in journal names and formats, botanical names are not in italic form, page numbers, issues etc. (see highlighted text with yellowish mark).

Annotated reviews are not available for download in order to protect the identity of reviewers who chose to remain anonymous.

Reviewer 3 ·

Basic reporting

The structure of the manuscript is complete and the research is driven by a clear hypothesis. The writing of the text is fairly good, although some grammer mistakes could be found occasionally.

Experimental design

Research question of the manuscript is well defined and meaningful. However, some key experiments were missing in the experimental design (see in the Additional comments).

Validity of the findings

Conclusions are well stated and statistics analyses were properly perfromed in the manuscript.

Additional comments

Photoperiod is a very critical environmental factor affecting plant growth and development, however, regulation of bermudagrass flowering by photoperiod is not clear. In this manuscript authored by Ji et al., the growth, nutrient absorption and flowering of common bermudagrass A12359 and African bermudagrass ABD11 under different day-time/light-time conditions were analyzed in detail. The results indicated that increase of day-time promoted the growth and accelerated the flowering of two bermudagrass plants, providing a possible method to accelerate breeding of this important turfgrass species. Although the manuscript has many merits, some major deficients should also be improved.

1. The flowering of plants, including bermudagrass, is correlated with accumulation temperature, which determine the time to reach the threshold of total carbon accumulation required for reproductive growth. Enlongation of light time also increase the total light energy illuminated on the plants. Thus, the observed accelerated flowering might be the result of increased light intensity rather than the change of light/dark period. To discriminate the meaning of photoperiod, I think the term "photoperiod/light period" should be changed to "light application time" throughout the manuscript.

2. What about the influence of light time change on seed number, seed size and seed viability? Since the aim of this study is to accelerate flowering, and the seed is the final target of flowering and cross breeding. The seed-related parameters thus must be provided in the manuscript.

3. Excess light is harmful to plants, especially to the reaction centers of photosystems in choloplasts. Leaf ROS level and photosynthesis-related parameters should be measured to evaluate whether the increase of light time is harmful or not to the photosystems.

4. Bermudagrass is a globally distributed grass species and different wild germplasms of bermudagrass have adapt to the local light intensity and other enviromental factors. As described by the authors, the growth and development of the two tested bermudagrass plants showed huge changes under different light times. Are the other germplasms behaved the same as these two test materials? Are they also flower early? If experiments using more materials cann't be performed, the possible responses of different germplasms should be discussed in discussion section.

5. Figure 5: The number of flowering (C and D) were not statistically analyzed as the number of branch (A and B).

6. line 213: "The long photoperiod was regularly longer than the short photoperiod, as seen by the branch numbers of A12359 and ABD11." what dose this sentence mean?

---

## Round 0.2 · Minor Revisions

Some remaining minor modifications are needed before accepting the MS

Reviewer 2 ·

Basic reporting

The manuscript has been improved significantly according to the reviewers comments. The manuscript can be accepted for publication after minor revisions as described below:

L33: N and P content
L38: earlier than
L257: Please use the unit of DLI as mol m-2 d-1 and also in the figure 5.
L340: nutrient content
L563: Figure 1 title is not complete. Please complete the sentence
Suggested title for Figure 2 and onwards: Plant height (cm), fresh weight (g) and dry weight (g) of Common bermuda grass (A12359) and African Bermuda grass (ABD11) grown in three light application times (22/2, 18/6 and 14/10 hrs light/dark).
Figure 2: Space needed in Y axis titles of all graphs between parameter and parenthesis. Please increase the Y axis unit differences in the figures b, c, e and f.
Describe the meaning of lower-case letters in details. Like, different letters within the light application times indicate significant differences at …….
Figure 5 title: DLI instead of DIL
I would recommend authors to calculate the average PPFD and temperature for each treatments and mention somewhere in the methodology section. For this type of experiment, the amount of light intensity used for plants must be specified. Still it is really difficult to understand the amount of light and temperature received by the plants in three light regimes.
The manuscript need to be thoroughly checked for typing and grammatical errors as well as structures of the figures and tables.

Experimental design

Need minor changes with few information.

Validity of the findings

Satisfactory.

Additional comments

Mentioned above.

Reviewer 3 ·

Basic reporting

no comment

Experimental design

no comment

Validity of the findings

no comment

Additional comments

The authors made efforts to improve their work. Although some of my raised questions were not fixed, I feel the manuscript can be accepted in the current form.

---

## Round 0.3 · accepted · Accept

The revised version of the manuscript can be accepted for publication.

The Section Editor provided the following list of edits:

EDITS LINE NO: / BEFORE / AFTER / [COMMENTS]

LINE 25: / extended light application time have been / extension of light application time has been / [.]
LINE 34: / under 22/2 h / under the 22/2 h / [.]
LINE 36: / genotype, A12359 / genotype A12359 / [.]
LINE 37: / genotype, ABD11 / genotype ABD11 / [.]
LINE 38: / might provide / may provide / [.]
LINE 39: / insight / insights / [.]
LINE 43: / plants growth / plant growth / [.]
LINE 86: / of world’s / of the world’s / [.]
LINE 101: / outstanding kinds. / outstanding germplasm. / [a cleaner finish]
LINE 109: / group and named / group named / [.]
LINE 184: / before got stabilized / before it stabilized / [.]
LINE 188: / bermudagrass, ABD11 did, / bermudagrass ABD11 did, / [.]
LINE 191: / measured in every / measured every / [.]
LINE 220: / And / And for / [.]
LINE 221: / at longer / at a longer / [.]
LINE 222: / their maximum / its maximum / [.]
LINE 222: / week 12t h. / week 12. / [.]
LINE 225: / at longer / at the longer / [.]
LINE 226: / reached their / reached its / [.]
LINE 227: / week 8t h / week 8 / [.]
LINE 230: / to shorter / to the shorter time / [.]
LINE 239: / week 12t h, / week 12, / [.]
LINE 294: / it has found / it was found / [.]
LINE 301: / timothy / ‘Timothy’ / [.]
LINE 305: / reproductive. / reproductive stages. / [.]